# Differentiable Causal Discovery Under Latent Interventions

**Gonçalo R. A. Faria**[*†]                                    GONCALORAFARIA@TECNICO.ULISBOA.PT

**André F. T. Martins** [*†‡]                                 ANDRE.T.MARTINS@TECNICO.ULISBOA.PT

**Mário A. T. Figueiredo** [*†]                              MARIO.FIGUEIREDO@TECNICO.ULISBOA.PT

[*] *Instituto Superior Técnico & LUMLIS (Lisbon ELLIS Unit), Universidade de Lisboa, Portugal.*

[†] *Instituto de Telecomunicações, Lisboa, Portugal.*

[‡] *Unbabel, Lisboa, Portugal.*

**Editors:** Bernhard Schölkopf, Caroline Uhler and Kun Zhang

## Abstract

Recent work has shown promising results in causal discovery by leveraging interventional data with gradient-based methods, even if the intervened variables are unknown. That previous work, however, assumes that the correspondence between samples and interventions is known, which is often unrealistic. We consider a scenario in which there is an underlying observational distribution that undergoes multiple interventions, but without knowledge of which intervention (if any) corresponds to each sample, and of how the interventions affect the system; *i.e.*, the interventions are entirely latent. To address this scenario, we propose a method based on neural networks and variational inference, by framing the problem as that of learning a shared causal graph among an infinite mixture (under a Dirichlet process prior) of intervention structural causal models. Experiments with synthetic and real data show that our approach and its semi-supervised variant are able to discover causal relations in this challenging scenario.

**Keywords:** causal discovery, latent interventions, variational inference, Dirichlet processes.

## 1. Introduction

Discovering causal relations among variables has countless applications in many scientific and data analysis fields (Peters et al., 2017). However, causal graphs are notoriously hard to learn from data: only strong assumptions can ensure that the correct causal structures can be recovered from observations alone, even with an unlimited supply of data. Recent work by Brouillard et al. (2020) has shown very promising results in causal discovery by leveraging interventional data with gradient-based learning, even if which variables are intervened (targeted) in each intervention is unknown. However, that work assumes that the *correspondence* between samples and interventions is known, which is often an unrealistic assumption. Our work addresses this limitation by proposing a method for causal discovery under *fully latent* interventions, through a neural-based variational approach that infers the correspondences between data samples and interventions. Our framework falls into the class of continuous constrained optimization methods for finding a DAG (*directed acyclic graph*); other approaches include constraint-based (Spirtes et al., 1993; Pearl, 2000; Spirtes et al., 2013) and score-based methods (Bouckaert, 1993; Heckerman et al., 1994; Chickering and Heckerman, 1997; Kuipers et al., 2021). We assume (following Brouillard et al. (2020)) that the data is clustered into intervention groups, but we relax their assumption that the correspondence between intervention groups and samples is known, opening the door to more realistic scenarios. We model these *latent* interventions as being generated by a *Dirichlet process prior* (DPP) and formulate the problem as one of end-to-end maximization of an *evidence lower bound* (ELBO). Our contributions are summarized as follows:

- We formulate causal discovery under latent interventions as searching for the shared causal graph among an infinite mixture of intervened structural causal models.

- We propose a variational formulation with a DPP to model this infinite mixture. We use latent intervention embeddings with shared parameters to model an unlimited number of interventions.

- We develop a semi-supervised variant of our method; for when we know the correspondence for a subset of the samples.

Our experiments on both synthetic and real data show that our method is able to discover causal structures, outperforming several baselines and reaching similar performance to methods that have full knowledge of the correspondence between samples and interventions. Although by no means an exhaustive experimental evaluation, our results are very promising and support the conclusion that the proposed method extends the range of scenarios in which causal discovery can be performed.

## 2. Background

### 2.1. Structural causal models

We assume a structured causal model (SCM) $\mathcal{M} := (S, p(\mathcal{E}))$ over $d$ *endogenous* variables $X = \{x_1, \ldots, x_d\}$, associated to $d$ independent *exogenous* variables $\mathcal{E} = \{\varepsilon_1, \ldots, \varepsilon_d\}$, functions $\mathcal{F} = \{\zeta_1, \ldots, \zeta_d\}$, and a collection $S$ of $d$ assignments,

$$x_j := \zeta_j(\mathbf{PA}_j^{\mathcal{G}}, \varepsilon_j), \qquad j = 1, \ldots, d, \tag{1}$$

where $\mathbf{PA}_j^{\mathcal{G}} \subseteq X \backslash \{x_j\}$ is the set of *parents* of $x_j$ according to a *directed acyclic graph* (DAG) $\mathcal{G}$, *i.e.*, $x_i \in \mathbf{PA}_j^{\mathcal{G}}$ if and only if there is a directed edge $x_i \to x_j$ in $\mathcal{G}$. An SCM $\mathcal{M}$ defines a unique joint distribution for $X$, usually referred to as the *entailed* distribution $p_{\mathcal{M}}(X)$, which can be factorized as

$$p_{\mathcal{M}}(X) = \prod_{j=1}^{d} p_{\mathcal{M}}(x_j | \mathbf{PA}_j^{\mathcal{G}}), \tag{2}$$

where $p_{\mathcal{M}}(x_j | \mathbf{PA}_j^{\mathcal{G}})$ is the conditional distribution of $x_j$, given its parents (Spirtes et al., 1993).

### 2.2. Interventions

Given an SCM $\mathcal{M}$, we obtain an *interventioned* SCM $\tilde{\mathcal{M}}$ by replacing one (or more) of the original assignments. Let $I$ be the set of variables targeted by the intervention; if $I = \emptyset$, $\tilde{\mathcal{M}} = \mathcal{M}$. For each variable $j \in I$, the intervention consists in one or more of the following actions: replacing the assignment function $\zeta_j$ by $\tilde{\zeta}_j$; replacing the parents $\mathbf{PA}_j^{\mathcal{G}}$ by a subset $\tilde{\mathbf{PA}}_j^{\mathcal{G}}$; changing the exogenous (noise) variable from $\varepsilon_j$ to $\tilde{\varepsilon}_j$. The SCM $\tilde{\mathcal{M}}$ generally has a different entailed distribution, called the *intervention* distribution:

$$p_{\tilde{\mathcal{M}}}(X) = \prod_{j \in \{1,..,d\} \backslash I} p_{\mathcal{M}}(x_j | \mathbf{PA}_j^{\mathcal{G}}) \prod_{j \in I} p_{\mathcal{M}; \mathrm{do}\left(x_j := \tilde{\zeta}_j(\tilde{\mathbf{PA}}_j^{\mathcal{G}}, \tilde{\varepsilon}_j)\right)}(x_j | \mathbf{PA}_j^{\mathcal{G}}). \tag{3}$$

If there are $K$ possible interventions, we denote the corresponding sets of target variables as $I^{(k)}$, for $k = 1, \ldots, K$, and the corresponding SCMs by $\tilde{\mathcal{M}}^{(k)}$.

We divide the types of interventions into: *atomic*, if the target variable $x_j$ is set to a constant value (*i.e.*, function $\tilde{\zeta}_j$ is constant); *stochastic*, if $x_j$ is set to a random variable $\tilde{\varepsilon}_j$ (*i.e.*, if $\tilde{\zeta}_j(\tilde{\mathbf{PA}}_j^{\mathcal{G}}, \tilde{\varepsilon}_j) = \tilde{\varepsilon}_j$); *imperfect* (or *soft*), if the interventioned assignment is changed and it still depends on a nonempty subset of $\mathbf{PA}_j^{\mathcal{G}}$, *i.e.*, $\tilde{\mathbf{PA}}_j^{\mathcal{G}} \subseteq \mathbf{PA}_j^{\mathcal{G}}$ and $\tilde{\mathbf{PA}}_j^{\mathcal{G}} \neq \emptyset$. We do not consider interventions that are able to add new elements to $\mathbf{PA}_j^{\mathcal{G}}$. This means that the intervention graph only differs from the observational by the removal of edges.

### 2.3. Faithfulness and Markov equivalence classes

Given a set $\mathcal{F} = \{\zeta_1, \ldots, \zeta_d\}$, where each function $\zeta_i$ is *sufficiently* dependent on all the arguments in $\mathbf{PA}_i^{\mathcal{G}}$, we obtain an SCM $\mathcal{M}$ whose computations strictly follow the structure of $\mathcal{G}$. In this scenario, $\mathcal{G}$ and $p_{\mathcal{M}}(X)$ are said to be mutually *faithful* since $\mathcal{G}$ encodes all and only the conditional independencies that hold in the entailed distribution. The set of faithful graphs that could entail a particular joint distribution is called the *Markov equivalence class* (MEC) (Verma and Pearl, 2013). If there is access to intervention data (in a set of interventions $\mathcal{I}$), it is possible to shrink the MEC to the so-called $\mathcal{I}$-MEC (Hauser and Bühlmann, 2012): the subset of graphs in the MEC that have the same conditional independencies after applying the interventions in $\mathcal{I}$.

### 2.4. Continuous constrained optimization for structure learning

Our work builds on a recent line of research that uses continuous constrained optimization to address causal discovery, initiated by Zheng et al. (2018) and extended by Brouillard et al. (2020) to cases where there is data from intervention distributions. In general, these approaches adopt the *maximum a posteriori* (MAP) criterion (a.k.a. penalized maximum likelihood). Based on a generative/sampling model $p(\mathcal{D}|\mathcal{G}, \theta)$ for data $\mathcal{D}$, given the graph structure $\mathcal{G}$ and parameters $\theta$, and on a *prior* $p(\mathcal{G})$ over graphs, the graph estimate is sought by maximizing the score function

$$\mathcal{S}(\mathcal{G}) := \max_{\theta} \log p(\mathcal{D}|\mathcal{G}, \theta) + \log p(\mathcal{G}). \tag{4}$$

If $\mathcal{D}$ is a collection of i.i.d. observations, then $p(\mathcal{D}|\mathcal{G}, \theta) = \prod_{i=1}^{n} p(x_i|\mathcal{G}, \theta)$. The prior $p(\mathcal{G})$ penalizes graph complexity to avoid over-fitting, with a typical choice being $p(\mathcal{G}) \propto \exp(-\xi|\mathcal{G}|)$, for $\xi > 0$ and where $|\mathcal{G}|$ is some graph complexity measure (*e.g.*, number of edges). With finite data, exact independence seldom occurs, thus graphs maximizing $\log p(\mathcal{D}|\mathcal{G}, \theta)$ alone would almost always be fully connected.

Central to this class of methods is the weighted adjacency matrix $W^{\mathcal{G}} \in \mathbb{R}_+^{d \times d}$, where $W_{ij}^{\mathcal{G}} > 0$ is equivalent to $(i, j) \in \mathcal{G}$, where $\mathcal{G}$ is treated as a parameter itself or as a function of the parameters. To ensure the estimated graph is a DAG, Zheng et al. (2018) proposed the constraint

$$\text{trace}\left(e^{W^{\mathcal{G}} \odot W^{\mathcal{G}}}\right) - d = 0, \tag{5}$$

where $e^A$ denotes the matrix exponential of matrix $A$ and $\odot$ is the Hadamard product. Several other methods apply non-linear models such as neural networks (Lachapelle et al., 2019; Zheng et al., 2020) and define $W^{\mathcal{G}}$ differently.

The works by Ng et al. (2020), Kalainathan et al. (2020), and Brouillard et al. (2020) treat the adjacency matrix as a random variable and relax the score in Eq. (4) to an argument $\Lambda \in \mathbb{R}^{d \times d}$,

$$\mathcal{S}^{\star}(\Lambda) := \max_{\theta} \mathbb{E}_{\mathcal{G} \sim \text{Bern}\left(\mathcal{G}; \sigma(\Lambda)\right)}\left[ \log p(\mathcal{D}|\mathcal{G}, \theta) + \log p(\mathcal{G}) \right], \tag{6}$$

where $\sigma(\Lambda)$ is the element-wise application of the sigmoid function $\sigma(u) = \exp(u)/(1 + \exp(u))$, and $\mathrm{Bern}\big(\mathcal{G}; \sigma(\Lambda)\big)$ is a distribution over graphs, with mutually independent edges, with expected value $\sigma(\Lambda)$. This score tends asymptotically to $\mathcal{S}(\mathcal{G})$ as $\sigma(\Lambda)$ progressively concentrates its mass on a single DAG $\mathcal{G}$.

## 3. Differentiable causal discovery under latent interventions

In this section, we present a score for perfect or imperfect fully latent interventions, and show how this score can be approximately maximized by using an efficient variational optimization algorithm.

### 3.1. Mixture of intervention distributions

We assume that the dataset $\mathcal{D}$ is produced by a mixture of SCMs, each resulting from an intervention applied to a base SCM $\mathcal{M}$. More specifically, $\mathcal{D}$ is partitioned into $K + 1$ exchangeable groups, with group $k$ containing i.i.d. samples from the intervention SCM $\tilde{\mathcal{M}}^{(k)}$ resulting from applying the $k^{\text{th}}$ intervention to the base SCM $\mathcal{M}$; the index $k = 0$ indicates the absence of intervention, *i.e.*, $\tilde{\mathcal{M}}^{(0)} = \mathcal{M}$ (observational model). We denote by $\tilde{\mathcal{M}} = \{\tilde{\mathcal{M}}^{(0)}, ..., \tilde{\mathcal{M}}^{(K)}\}$ the ensemble of SCMs.

The latent variables $z^{(i)} \in \{0, ..., K\}$ indicate which SCM generated the $i$-th sample, with $z^{(i)} = k$ meaning that $x^{(i)}$ is a sample of the SCM $\tilde{\mathcal{M}}^{(k)}$. Treating these correspondences $z^{(i)}$ as latent is a distinctive aspect of our work; while Brouillard et al. (2020) also assume unknown $\tilde{\mathcal{M}}$, they assume that $z^{(i)}$ is observed. We call the scenario where both $\tilde{\mathcal{M}}$ and $z^{(i)}$ are unknown as *fully latent interventions*. Marginalizing with respect to the latent $z^{(i)}$ yields the mixture model

$$p(x^{(i)}|\tilde{\mathcal{M}}) = \sum_{k=0}^{K} \underbrace{p(z^{(i)} = k)}_{\tau_k} \, p\big(x^{(i)}|z^{(i)} = k, \tilde{\mathcal{M}}\big) = \sum_{k=0}^{K} \tau_k \, p_{\tilde{\mathcal{M}}^{(k)}}(x^{(i)}). \tag{7}$$

Conditioning on $z^{(i)}$ and invoking Eq. (3) leads to

$$p(x^{(i)}|z^{(i)}, \tilde{\mathcal{M}}) = \sum_{k=0}^{K} \mathbb{I}(z^{(i)} = k) \prod_{j \in \{1,..,d\} \setminus I^{(k)}} p_{\mathcal{M}}(x_j^{(i)}|\mathbf{PA}_j^{\mathcal{G}}) \prod_{j \in I^{(k)}} p_{\mathcal{M};do\big(x_j := \tilde{\zeta}_j^k(\tilde{\mathbf{PA}}_j^{\mathcal{G}}, \tilde{\varepsilon}_j)\big)}(x_j^{(i)}|\mathbf{PA}_j^{\mathcal{G}}).$$

We also consider that, like the group memberships, the set of targets $I^{(k)}$ of each intervention is unknown, except for $I^{(0)} = \emptyset$.

### 3.2. Distribution over causal graphs

We represent the causal graph $\mathcal{G}$ via the adjacency matrix $A^{\mathcal{G}} \in \{0, 1\}^{d \times d}$. Following previous work, our prior models each entry $A_{ij}^{\mathcal{G}}$, corresponding to edge $x_i \rightarrow x_j$, as a Bernoulli variable independent of all the others,

$$p(\mathcal{G}) = \prod_{i,j=1}^{d} \sigma(\xi_{ij})^{A_{ij}^{\mathcal{G}}} \big(1 - \sigma(\xi_{ij})\big)^{1 - A_{ij}^{\mathcal{G}}}, \tag{8}$$

where the $\xi_{ij}$ are hyper-parameters. In this paper, we set $\xi_{ij} = \xi_{\mathcal{G}}$, for all $i, j$; however, in practice, a domain expert using the proposed method can embed prior knowledge in these hyper-parameters

(our method can be straightforwardly adapted to that case). This prior over graphs is simplistic since it does not encode that $\mathcal{G}$ has to be a DAG.

The adoption of a probabilistic prior $p(\mathcal{G})$ should not be seen as expressing that it is in fact a random object, but rather as a subjective prior in the context of epistemic uncertainty about it.

### 3.3. Intervention embeddings and shared intervention space

We use density estimators, *e.g.*, neural networks and normalizing flows (Rezende and Mohamed, 2015), to model the conditional densities in both the observational and interventional distributions. With this goal in mind, we use an appropriate encoding of the changes in the intervened assignments and the intervention targets. The set of targets in the $k$-th intervention $\tilde{\mathcal{M}}^{(k)}$ is indicated by a $d$-dimensional binary vector $r_k = [r_{k1}, \ldots, r_{kd}] \in \{0, 1\}^d$, with $r_{kj} = 1 \Leftrightarrow j \in I^{(k)}$. Since $I^{(0)}$ has no targets (it corresponds to the observational SCM $\mathcal{M}$), we have $r_0 = [0, \ldots, 0]$. To encode the type of intervention, we introduce the *intervention embedding vector* $u_k \in \mathbb{R}^h$, where $h$ is a hyper-parameter. Each vector $u_k$ represents the changes in the affected assignments for intervention $\tilde{\mathcal{M}}^{(k)}$, in a way that will become clear in the next paragraph. We denote by $\mathcal{R} = [r_0, r_1, \ldots, r_K] \in \{0, 1\}^{(K+1) \times d}$ the matrix of intervention target indicators, and by $\mathcal{U} = [u_0, u_1, \ldots, u_K] \in \mathbb{R}^{(K+1) \times h}$ the matrix of intervention embeddings. The pair $(\mathcal{R}, \mathcal{U})$ represents the set of interventions $\tilde{\mathcal{M}}$.

Putting everything together, when given the graph $\mathcal{G}$, interventions $\tilde{\mathcal{M}}$, indicator $z$, and assuming the intervention is imperfect, the conditional log-probability of single data-point $x$ is given by

$$
\log p(x|z, \tilde{\mathcal{M}}, \mathcal{G}; \theta) =
$$
$$
= \sum_{k=0}^{K} \mathbb{I}(z = k) \sum_{j=1}^{d} (1 - r_{kj}) \log \underbrace{g_j(x_j | A_j^{\mathcal{G}} \odot x, u_0; \theta_j)}_{=: \, p_{\mathcal{M}}(x_j | \mathbf{PA}_j^{\mathcal{G}})} + r_{kj} \log \underbrace{g_j(x_j | A_j^{\mathcal{G}} \odot x, u_k; \theta_j)}_{=: \, p_{\tilde{\mathcal{M}}^{(k)}}(x_j | \mathbf{PA}_j^{\mathcal{G}})}
$$
$$
= \sum_{j=1}^{d} \log g_j(x_j | A_j^{\mathcal{G}} \odot x, (e_z^\top \mathcal{R})_j \, (e_z^\top \mathcal{U} - u_0) + u_0; \theta_j), \tag{9}
$$

where $e_z \in \mathbb{R}^{K+1}$ is a one-hot vector representation of $z$, $A_j^{\mathcal{G}}$ is the $j$-th row of the adjacency matrix $A^{\mathcal{G}}$, thus $A_j^{\mathcal{G}} \odot x$ corresponds to selecting the entries of $x$ in $\mathbf{PA}_j^{\mathcal{G}}$. The conditional densities $g_1, \ldots, g_d$ are parameterized by $\theta = (\theta_1, \ldots, \theta_d)$. We will consider below several forms for these conditional densities, *e.g.*, using parametric families and normalizing flows. Crucially, each $g_j$ is a conditional density of $x_j$, and the parameters $\theta_j$ are *shared* by all the interventions—only the intervention-specific embedding vector $u_k$ changes depending on the intervention, as explained below. This enables dealing with an unlimited number of interventions, as we shall see.

### 3.4. Modeling the conditional densities

A simple nonlinear model for the conditional densities $g_j$ can be constructed using neural networks. We use a neural network $\text{NN}([u_k, A_j^{\mathcal{G}} \odot x]; \theta_j) : \mathbb{R}^{(h+d)} \to \mathbb{R}^m$, a non-linear mapping parameterized by $\theta_j$, that receives the concatenation of the parents of $x_j$ and the intervention embedding $u_k$ and outputs the $m$ parameters of some distribution $f(x_j; \text{NN}([u_k, A_j^{\mathcal{G}} \odot x]; \theta_j))$ for variable $x_j$. There are many possible choices for the distribution $f$, depending on the problem at hand and on whether $x_j$ is discrete or continuous; for example, Poisson ($m = 1$), Bernoulli ($m = 1$), univariate Gaussian

($m = 2$), categorical ($m$ is the number of categories minus one). In this paper, we focus on three density families in $\mathbb{R}$, with which we experiment in Section 4.

**Linear Gaussian:** in this case, we use a neural network $\text{NN}(u_k; \theta_j)$ to output coefficients $\tilde{a}_j \in \mathbb{R}^d$, and $\tilde{\sigma}_j, \tilde{b}_j \in \mathbb{R}$. Then, we use these as parameters of a Gaussian distribution whose mean is an affine transformation of the values of the parents of $x_j$:

$$g_j(x_j | A_j^{\mathcal{G}} \odot x, u_k) = \mathcal{N}\big(\tilde{a}_j^\top \big(A_j^{\mathcal{G}} \odot x\big) + \tilde{b}_j, \tilde{\sigma}_j^2\big).$$

**Non-Linear Gaussian:** this model uses a neural network $\text{NN}([u_k, A_j^{\mathcal{G}} \odot x]; \theta_j)$ to output coefficients $\tilde{\mu}_j \in \mathbb{R}$ and $\tilde{\sigma}_j \in \mathbb{R}$, given the values of the parents of $x_j$ as input. Then, we use these as parameters of a Gaussian distribution:

$$g_j(x_j | A_j^{\mathcal{G}} \odot x, u_k) = \mathcal{N}\big(\tilde{\mu}_j, \tilde{\sigma}_j^2\big).$$

**Normalizing flows:** to model non-linear non-Gaussian conditional densities, we use normalizing flows (Rezende and Mohamed, 2015), which transform a base probability density (Gaussian, in our case) through a sequence of invertible mappings $\tau(x_j; \tilde{W}_j) = \tau_l \circ \tau_{l-1} \cdots \circ \tau_1(x_j; \omega_1)$, where $\tilde{W}_j = \{\omega_1, \ldots, \omega_l\}$. We use a model introduced by Huang et al. (2018), called *deep sigmoidal flows* (DSF), where each of the invertible mappings has the form

$$\tau_l(x) = \sigma^{-1}(w_l^\top \sigma(a_l x + b_l)), \qquad w_l \in \Delta_{F-1}, \quad a_l \in \mathbb{R}_+^F, \quad b_l \in \mathbb{R}^F,$$

where $\Delta_{F-1}$ is the probability simplex and $F$ is a hyper-parameter. We use a neural network $\text{NN}([u_k, A_j^{\mathcal{G}} \odot x]; \theta_j)$ to output the parameters $\tilde{W}_j$. With the former, we obtain a controllable flow $\tau(x_j; \tilde{W}_j)$ that, given $x_j$, outputs the parameters of a Gaussian distribution $\tilde{\mu}, \tilde{\sigma}$. Altogether, the joint density has the following form:

$$g_j(x_j | A_j^{\mathcal{G}} \odot x, u_k) = \left| \det \left( \frac{\partial \tau(x_j; \tilde{W}_j)}{\partial x_j} \right) \right| \mathcal{N}\big(\tilde{\mu}, \tilde{\sigma}^2\big).$$

### 3.5. Modeling latent interventions with a Dirichlet process

To obtain a complete statistical description of the data-generating process, we still need a prior distribution for the latent interventions, apart from the sampling model $g_j$. Namely, we need a prior distribution for the correspondence $z$, the intervention embeddings $\mathcal{U}$, and the intervention targets $\mathcal{R}$. Furthermore, while experimentally we can have scenarios where $K$, the number of latent interventions, is known, in general, given a data set, it may not be clear what the number of latent interventions is. Therefore, we will formulate the model with a potentially unspecified $K$. We do this by using a *Dirichlet process prior* (DPP) with a stick-breaking representation (Ferguson, 1973; Sethuraman, 1991) as the prior distribution for the variables associated with the latent interventions.

The generative *story* underlying the prior is as follows. We first draw the graph $\mathcal{G}$ from $p(\mathcal{G})$, as described in Eq. (8). Then, for $k = 0, 1, \ldots$, we sample the variables $u_k$ and $r_k$, associated with each intervention $\tilde{\mathcal{M}}^{(k)}$, as well as the probability $\beta_k$ of picking that intervention as a stick-breaking process, with scaling parameter $\alpha > 0$ (which controls the clustering effect of the DPP)

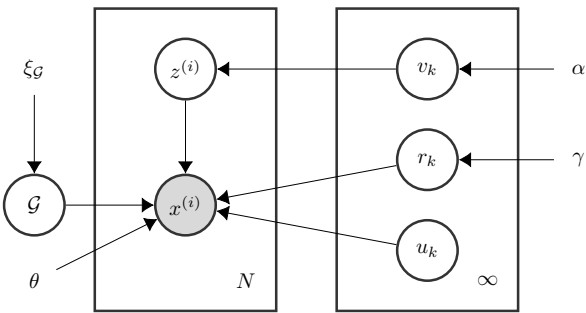

Figure 1: Graphical model representation of the Dirichlet Process Mixture model augmented for the causal discovery problem.

and hyperparameter $\gamma$ (which controls the sparsity of the intervention targets), as follows:

$$
\begin{aligned}
u_k &\sim \mathcal{N}(0, I_h), \qquad r_{kj} \sim \text{Bern}(\sigma(\gamma)), \quad j = 1, \ldots, d, \\
\beta_k &= v_k \prod_{k'=0}^{k-1} (1 - v_{k'}), \quad \text{with } v_k \sim \text{Beta}(1, \alpha).
\end{aligned} \tag{10}
$$

Then, to generate the data, we first sample the intervention index $z^{(i)}$, and then sample a point $x^{(i)}$ conditioning on the corresponding intervention $\tilde{\mathcal{M}}^{(z^{(i)})}$:

$$
\begin{aligned}
z^{(i)} &\sim \text{Cat}(\beta_0, \beta_1, \ldots, \beta_k, \ldots), \\
x_j^{(i)} &\sim \begin{cases} g_j(x_j | A_j^{\mathcal{G}} \odot x^{(i)}, u_0) & \text{if } r_{z^{(i)}j} = 0 \\ g_j(x_j | A_j^{\mathcal{G}} \odot x^{(i)}, u_{z^{(i)}}) & \text{if } r_{z^{(i)}j} = 1. \end{cases}
\end{aligned} \tag{11}
$$

Figure 1 contains a graphical model representation of this joint distribution.

### 3.6. Variational approximation

To obtain the marginal likelihood $p(\mathcal{D}|\mathcal{G}; \theta)$ to be used in Eq. (4), it is necessary to marginalize over the joint distribution of the model present in Section 3.5, *i.e.*,

$$
\log p(\mathcal{D}|\mathcal{G}; \theta) = \sum_{i=1}^{N} \log p(x^{(i)}|\mathcal{G}; \theta) = \sum_{i=1}^{N} \log \mathbb{E}_{z^{(i)}, \tilde{\mathcal{M}} \sim p(z^{(i)}, \tilde{\mathcal{M}})} \big[ p(x^{(i)}|z^{(i)}, \tilde{\mathcal{M}}, \mathcal{G}; \theta) \big]. \tag{12}
$$

However, the exact maximization of this marginal log-likelihood (which involves a product of Gaussians, Beta distributions, and complex conditional distributions generated by neural networks) is intractable. Therefore, we resort to *approximate variational inference* (Blei et al., 2017), based on a family of tractable variational distributions $q_\phi(z^{(i)}, \tilde{\mathcal{M}})$ to approximate the true posterior $p(z^{(i)}, \tilde{\mathcal{M}}|x^{(i)}, \mathcal{G}; \theta)$. For the variables associated with latent interventions $\tilde{\mathcal{M}}$, we propose the fully factorized and finite variational family $\mathcal{Q}$

$$
q(\tilde{\mathcal{M}}) = \prod_{k=0}^{K} \Big( \prod_{j=1}^{d} q_R(r_{kj}) \Big) \Big( \prod_{l=1}^{h} q_U(u_{kl}) \Big) q_V(v_k), \tag{13}
$$

where the hyper-parameter $K$ defines the truncation level of the variational approximation, and the variational marginals associated with $v_k$, $u_{kl}$, and $r_{kj}$ take the following forms:

$$
\begin{aligned}
q_V(v_k; \rho_k, w_k) &= \text{Beta}\big(\rho_k w_k, (1 - \rho_k) w_k\big), \\
q_U(u_{kl}; \mu_{kl}, \sigma_{kl}) &= \mathcal{N}(\mu_{kl}, \sigma_{kl}^2), \\
q_R(r_{kj}; \pi_{kj}) &= \text{Bern}(\pi_{kj}),
\end{aligned}
\tag{14}
$$

where $\pi_{kj}, \mu_{kl}, \sigma_{kl}, \rho_k$, and $w_k$ are free parameters to be optimized, for each $k \in \{0, \ldots, K\}$, $l \in \{1, \ldots, h\}$, and $j \in \{1, \ldots, d\}$. For the distribution of interventional assignments $z$, we propose the following variational posterior:

$$
q_Z(z) \propto \exp\left( \frac{u_z^\top \text{NN}(x; \phi_Z)}{\sqrt{h}} \right), \qquad k = 1, \ldots, K, \tag{15}
$$

where $\text{NN}(x; \phi_Z) : \mathbb{R}^d \to \mathbb{R}^h$ is a neural network. Appendix D provides details about the derivation and, in the case of the Beta distribution, closed-form approximation of the Kullback–Leibler divergence between the proposals and the prior. The architecture of the neural networks used in the experiments is described in Appendix F. We use the shorthand $\phi$ to denote the vector of all variational parameters, which includes $\phi_Z, \mu_{kl}, \sigma_{kl}, \rho_k, w_k, \pi_{kj}$ for all $k \in \{0, \ldots K\}, l \in \{1, \ldots, h\}$, and $j \in \{1, \ldots, d\}$.

The ingredients above yield the following lower bound for the marginal log-likelihood in Eq. (12):

$$
\log p(\mathcal{D}|\mathcal{G}; \theta) \geq \sum_{i=1}^N \underbrace{\mathbb{E}_{z^{(i)}, \tilde{\mathcal{M}} \sim q(z^{(i)}, \tilde{\mathcal{M}}; \phi)}\big[ \log p(x^{(i)} | z^{(i)}, \tilde{\mathcal{M}}, \mathcal{G}; \theta) \big] - D_{\text{KL}}\big[ q(z^{(i)}, \tilde{\mathcal{M}}) || p(z^{(i)}, \tilde{\mathcal{M}}) \big]}_{\text{ELBO}_{q(z^{(i)}, \tilde{\mathcal{M}}; \phi)}(x^{(i)}, \mathcal{G}; \theta)}.
$$

For different choices of $\phi$, we get different lower bound approximations to the marginal log-likelihood. By maximizing the ELBO w.r.t. $\phi$ we minimize the approximation gap, which equals the KL divergence between the approximate and the true posterior.

### 3.7. A score for latent interventions

Using the log-likelihood in Eq. (12), we write a new score function $\mathcal{S}(\mathcal{G})$ for our model with latent interventions, with an associated relaxation to support a weighted adjacency $\sigma(\Lambda)$, as shown in Eq. (6). As discussed in Section 3.6, in general, we will not be able to exactly maximize this score. However, using the variational approximation above, and associated variational family $\mathcal{Q}$, we can approximate the score $\mathcal{S}(\mathcal{G})$ with a surrogate score $\mathcal{S}^{\mathcal{Q}}$ as follows:

$$
\mathcal{S}(\Lambda) \geq \underbrace{\max_{\theta, \phi} \mathbb{E}_{\mathcal{G} \sim \text{Bern}\big(\mathcal{G}; \sigma(\Lambda)\big)} \Big[ \sum_{i=1}^N \text{ELBO}_{q(z^{(i)}, \tilde{\mathcal{M}}; \phi)}(x^{(i)}, \mathcal{G}; \theta) + \log p(\mathcal{G}) \Big]}_{\mathcal{S}^{\mathcal{Q}}(\Lambda)}. \tag{16}
$$

The gap between the relaxed score $\mathcal{S}(\Lambda)$ and our surrogate $\mathcal{S}^{\mathcal{Q}}(\Lambda)$ tends asymptotically to the KL divergence between the best approximate posterior from $\mathcal{Q}$ and the true posterior, as $\sigma(\Lambda)$ progressively concentrates its mass on single DAG $\mathcal{G}$; more concretely,

$$
\mathcal{S}(\Lambda) - \mathcal{S}^{\mathcal{Q}}(\Lambda) = \mathbb{E}_{\mathcal{G} \sim \text{Bern}\big(\mathcal{G}; \sigma(\Lambda)\big)} \Big[ D_{KL}\big[ q(z^{(i)}, \tilde{\mathcal{M}}; \phi^*) || p(z^{(i)}, \tilde{\mathcal{M}} | x^{(i)}, \mathcal{G}; \theta^*) \big] \Big], \tag{17}
$$

where $\theta^*$ and $\phi^*$ are, respectively, the model and variational parameters that maximize Eq. (16).

### 3.8. Inference algorithm

The surrogate score, coupled with the acyclicity constraint of Zheng et al. (2018), enables formulating causal discovery under latent interventions as the following optimization problem:

$$\Lambda^* = \arg\max_{\Lambda} \mathcal{S}^{\mathcal{Q}}(\Lambda), \quad \text{subject to} \quad \underbrace{\text{Tr}\left(e^{\sigma(\Lambda)}\right) - d = 0}_{h(\Lambda)}. \tag{18}$$

Following Zheng et al. (2018), we use the *augmented Lagrangian* (AL) method (Hestenes, 1969; Powell, 1969; Glowinski and Marroco, 1975) to address this problem via a sequence of unconstrained ones. Estimating the gradients using the path-wise gradient estimator (Kingma and Welling, 2014; Rezende et al., 2014), each unconstrained optimization subproblem reduces to sampling the graph and the latent variables from the variational posteriors using the reparametrization trick, minimizing the following objective:

$$\mathcal{L}(\theta, \phi, \Lambda, z, \tilde{\mathcal{M}}, \mathcal{G}; x, \mu_t, \varphi_t) = -\log p(x|z, \tilde{\mathcal{M}}, \mathcal{G}; \theta) + \Omega(\phi) - \xi_{\mathcal{G}}||\Lambda||_1 + \varphi_t h(\Lambda) + \frac{\mu_t}{2}h(\Lambda)^2,$$

where $\Omega(\phi)$ denotes the sum of the KL divergences, and $\mu_t$ and $\varphi_t$ are the AL parameters at iteration $t$. To estimate the gradients of $\Lambda$, $q(z)$, and $q(r_{kj})$, we use a Gumbel-softmax continuous relaxation (Jang et al., 2017; Maddison et al., 2017), which, for the causal graph's distribution, was combined with the straight-through gradient estimator (Bengio et al., 2013), to make sure the graph samples actually represent the hard dependencies of the SCM, instead of fractional ones. Having estimated the gradients, for each sample, we average them and feed them to the first order stochastic optimization algorithm Adam (Kingma and Ba, 2015). The method is implemented in PyTorch (Li et al., 2020) and the code is available at `github.com/goncalorafaria/causaldiscovery-latent-interventions`.

### 3.9. Semi-supervised extensions

The proposed causal discovery algorithm can be extended to cases where we have information about the correspondences $z^{(i)}$ and/or the intervention targets $I^{(k)}$. To achieve this, we only have to ignore the corresponding variational posteriors and use the observed values $z^{(i)}$ and $r^{(i)}$ as constants. We designate the original version of our model as **latent**, the one with observed $z^{(i)}$ as the **unknown** variant, and the one with observed $z^{(i)}$ and $r^{(i)}$ as **known**. It is also straightforward to extend our method to a *semi-supervised* setting, in which the correspondences $z^{(i)}$ are observed for a fraction of the samples. In this scenario, we still use the samples from the variational posterior to predict the unobserved $z^{(i)}$ and use the observed ones to improve the variational posterior. In essence, the semi-supervised variant interpolates between the **latent** and the **unknown** variants. Following Kingma et al. (2014), we extend the ELBO objective (in our case, the lower bound on $\log p(\mathcal{D}|\mathcal{G}; \theta)$). Letting $\mathcal{O}$ be the set of the indices for which the intervention assignment $z^{(i)}$ is known, we write the semi-supervised score $\mathcal{S}^{\text{SSL}}(\Lambda)$ as the surrogate score from Eq. (16), by replacing the ELBO with

$$\mathcal{L}^{\text{SSL}}(\mathcal{D}, \mathcal{G}) = \sum_{i=1}^{N} \text{ELBO}_{\theta, q(z^{(i)}, \tilde{\mathcal{M}})}(x^{(i)}, \mathcal{G}) + \kappa \, \mathbb{E}_{\tilde{\mathcal{M}} \sim q(\tilde{\mathcal{M}})}\left[\sum_{i \in \mathcal{O}} \log q(z^{(i)}|x^{(i)}, \tilde{\mathcal{M}}; \phi_z)\right].$$

In the first term, for $i \in \mathcal{O}$, rather than sampling from $q(z^{(i)}|x^{(i)}, \tilde{\mathcal{M}}; \phi_z)$, we use the observed value. Finally, $\kappa \in ]0, 1[$ is a hyper-parameter controlling the relative importance of the supervised component of the objective.

## 4. Experiments

We tested our method on synthetic and real data. Experiments on synthetic data allow for a systematic, controlled assessment of different methods in different scenarios (graph size and density, intervention and assignment types). To generate SCMs, intervene on them, and sample from the corresponding distributions, we created a Pytorch package available at `github.com/goncalorafaria/PyCausal`. Appendix E provides a detailed description of how we generate SCMs.

**Single-node interventions** : We generate SCMs with $d = 10$ variables with a Erdős-Rényi scheme (Erdős and Rényi, 1959) with expected number of edges per node $e \in \{1, 4\}$.

We generated 10 SCMs for each combination of $e$, conditional density (linear Gaussian, non-linear Gaussian, and normalizing flow), and intervention type (stochastic and imperfect). In each of the SCMs, we performed one intervention for every variable. For each SCM in each configuration, we explored the following hyperparameter range: $\xi_{\mathcal{G}} \in \{-.1, -.01, 0, .01, .1\}$, and $\gamma \in \{-.1, -.01\}$. For the remaining hyperparameters, we set $\alpha = 9$, $h = 248$, and the truncation level $K = 11$—the hyperparameter configuration that achieved the best log-likelihood on a validation set. From the generated SCM, we produced a dataset with $n = 10^4$ samples, where each intervention has $\lfloor \frac{n}{d+1} \rfloor$ elements. This dataset was split into training (80%) and validation (20%). The models were trained for 500 epochs for the first iteration of the augmented Lagrangian and 50 epochs for the remaining ones, with a full batch ($B = 8000$) until $h(\Lambda) < 10^{-8}$, and learning rate of $10^{-2.5}$.

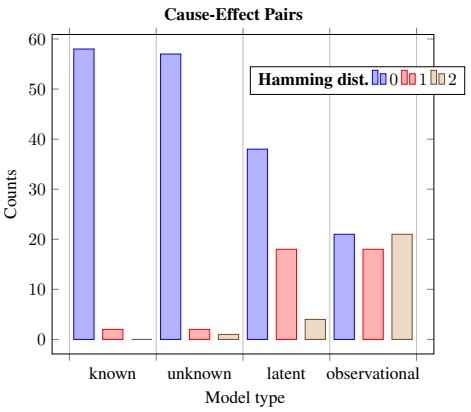

Figure 2: Histogram of Hamming distance in the experiments with cause-effect pairs.

The results of these experiment are shown in Table 1. The metric wed is *Hamming distance* (HD) between the adjacency matrices of the ground-truth graph and the estimated one. The columns in Table 1 correspond to the different variants described in Section 3.9. We additionally include a "naïve" observational model (a baseline which ignores the existence of intervention data, *i.e.*, that assumes $K = 0$ as if all data was generated by the observational model). The experimental results show that our method (the latent variant) consistently outperforms the observational baseline and is worse than the unknown variant, as expected since it uses less information, but only slightly. This indicates that taking into account latent interventions, when they are present, improves the recovery of the causal graph.

**Single-node interventions on cause-effect pairs:** We generated two-variable linear Gaussian SCMs (cause-effect pairs) with edge probability $e = \frac{2}{3}$ (equal probability for each of the three possible graphs). We sampled 60 SCMs, and generate a dataset of $n = 999$ samples, where each intervention has 333 elements. The sampled graph $\mathcal{G}$, with variables $A$ and $B$ is one of 3 possible graphs, $A$ causes $B$, $B$ causes $A$, or $A$ and $B$ have no cause-effect relation. We compare the variants presented in Section 3.9 in addition to the naïve observational baseline. Figure 2 contains an histogram of Hamming distances for each of the variants. Edges in the anti-causal direction cost HD = 2, missing edge or wrong edge is HD = 1, and correct graph is HD = 0. The results show

| Model Type | $e$ | latent | unknown | known | observational |
|---|---|---|---|---|---|
| | | *Stochastic Interventions:* | | | |
| Linear Gaussian | | $5.9 \pm 6.2$ | $3.4 \pm 3.2$ | $0.5 \pm 1.3$ | $10.3 \pm 7.8$ |
| Non-Linear Gaussian | 1 | $12.2 \pm 3.9$ | $10.3 \pm 2.5$ | $7.0 \pm 3.6$ | $13.7 \pm 3.8$ |
| Non-Linear Non-Gaussian | | $8.7 \pm 6.6$ | $8.0 \pm 2.7$ | $6.6 \pm 2.2$ | $11.3 \pm 5.0$ |
| Linear Gaussian | | $27.2 \pm 6.2$ | $24.1 \pm 5.8$ | $15.6 \pm 6.0$ | $39.6 \pm 5.0$ |
| Non-Linear Gaussian | 4 | $35.8 \pm 3.8$ | $30.3 \pm 5.3$ | $27.7 \pm 4.3$ | $37.5 \pm 5.2$ |
| Non-Linear Non-Gaussian | | $36.1 \pm 4.4$ | $35.5 \pm 8.1$ | $31.5 \pm 5.6$ | $40.2 \pm 6.9$ |
| | | *Imperfect Interventions:* | | | |
| Linear Gaussian | | $5.8 \pm 4.2$ | $6.2 \pm 3.06$ | $4.7 \pm 3.6$ | $10.4 \pm 2.9$ |
| Non-Linear Gaussian | 1 | $9.3 \pm 2.4$ | $8.9 \pm 2.5$ | $7.8 \pm 3.9$ | $10.5 \pm 2.8$ |
| Non-Linear Non-Gaussian | | $8.8 \pm 3.0$ | $9.1 \pm 3.5$ | $7.9 \pm 1.4$ | $11.5 \pm 5.4$ |
| Linear Gaussian | | $35.9 \pm 8.3$ | $29.7 \pm 5.6$ | $17.7 \pm 7.9$ | $39.1 \pm 9.1$ |
| Non-Linear Gaussian | 4 | $32.1 \pm 6.0$ | $32.6 \pm 5.8$ | $32.8 \pm 5.4$ | $39.8 \pm 9.3$ |
| Non-Linear Non-Gaussian | | $30.4 \pm 12.2$ | $30.2 \pm 11.2$ | $25.8 \pm 3.9$ | $36.7 \pm 9.8$ |

Table 1: Hamming distances on synthetic 10 variable SCMs.

for this simple problem that the causal graph cannot be identified from observational data alone, and that our method correctly identifies most of the cause-effect pairs, even without information about the intervention assignments.

**Real-world data:** Finally, we tested our method on the flow cytometry dataset of Sachs et al. (2005). Table 2 compares the estimated graph, under different conditional density assumptions and intervention types, to the consensus graph of Sachs et al. (2005). The results on this dataset show that our method outperforms several baselines, including methods that use information about the correspondences between interventions and targets, as well as FCI (Spirtes et al., 1993), which supports latent confounders (but not intervention data). Reasons that might justify the relatively good results on this standard problem include: (i) the causal sufficiency assumption may not hold, (ii) the interventions may not be as specific as stated, and (iii) the ground truth network is possibly not a DAG, since feedback loops are common in cellular signaling networks as noted by Mooij et al. (2020); Brouillard et al. (2020). These reasons can potentially be detrimental to the other methods, whereas our method appears to be robust to them.

## 5. Related Work

The previous work that is closest to ours in that by Brouillard et al. (2020). They assume, as we do, that the data is clustered in intervention groups, but we relax their assumption that the correspondence between intervention groups and samples is known, which is a more realistic assumption.

The so-called "known" and "unknown" variants of our method share the assumptions of Brouillard et al. (2020); however, in our method, the number of neural networks is independent on the number of interventions, which allows scaling to many interventions. We achieve this by encoding the assignments change associated with each intervention using a specific latent variable that we call intervention embedding and conditioning a shared model on it when computing the log-probability.

|  | HD | tp | fn | fp | rev | $F_1$ score |
|---|---|---|---|---|---|---|
| GIES (Hauser and Bühlmann, 2012) | 38 | 10 | 0 | 41 | 7 | 0.33 |
| CAM (Bühlmann et al., 2014) | 35 | 12 | 1 | 30 | 4 | 0.51 |
| IGSP (Wang et al., 2017) | 18 | 4 | 6 | 5 | 7 | 0.42 |
| DCDI-G (Brouillard et al., 2020) | 36 | 6 | 2 | 25 | 9 | 0.31 |
| DCDI-DSF (Brouillard et al., 2020) | 33 | 6 | 2 | 22 | 9 | 0.33 |
| FCI (Spirtes et al., 1993) | 35 | 4 | 12 | 21 | 5 | 0.22 |
| Imperfect Linear Gaussian (**ours**) | 33 | 7 | 11 | 22 | 3 | 0.30 |
| Imperfect Non-Linear Gaussian (**ours**) | 19 | 7 | 11 | 8 | 0 | 0.42 |
| Imperfect Normalizing Flow (**ours**) | 30 | 9 | 9 | 21 | 1 | 0.38 |
| Perfect Linear Gaussian (**ours**) | 23 | 8 | 10 | 13 | 3 | 0.41 |
| Perfect Non-Linear Gaussian (**ours**) | 24 | 11 | 7 | 17 | 1 | 0.48 |
| Perfect Normalizing Flow (**ours**) | 23 | 7 | 11 | 12 | 2 | 0.38 |

Table 2: Results for the flow cytometry dataset. The results for the baselines are reproduced from the work of Brouillard et al. (2020), with the exception of FCI.

There are many works that do causal discovery with latent confounders (Spirtes et al., 1993; Colombo et al., 2012; Claassen et al., 2013; Ogarrio et al., 2016). We view those works as complementary to ours, since in many situations the influence of latent confounding can render unreliable methods that assume causal sufficiency, such as ours. The scenario of causal discovery with latent confounders and ours are distinct in terms of underlying assumptions. We propose a scenario where interventions exists and they are fully latent: *i.e.*, we do not know what they alter or which samples are affected. While, we could regard these interventions as variables in a *expanded* causal model, hence latent confounders in that causal model, treating them in that way would disregard the structure of the problem, which is key to the success of our approach (looking for this structure is what enables discovering causal-effect relations). In practice, this leads to inferior results, as the results from Table 2 with the FCI (Spirtes et al., 1993) algorithm show.

Our method can be seen as a *variational autoencoder* (VAE) with a DPP, with a stick-breaking representation. The work by Nalisnick and Smyth (2017) introduces a VAE where the stick-breaking weights are the latent variables. Our model differs from theirs in that we use the entire DP (including the atoms). We only use the stick-breaking weights in the KL divergence of the correspondence variable $z$. This KL divergence has a closed-form, so we do not sample the stick-breaking weights as they do. van den Oord et al. (2017) proposes a VQ-VAE model with discrete latent variables, each represented as a latent embedding vector (an atom). Our approach is similar in that we use atoms (the intervention embeddings and intervention targets) to represent discrete latent variables. However, we do it in a statistically sounder way that more naturally fits our application.

## 6. Conclusion

We introduced an efficient variational optimization algorithm for causal structure learning under latent interventions. Our results are competitive with other state-of-the-art algorithms on the flow cytometry data. On synthetic data, our approach recovers causal relations even in our most challenging scenario, and it consistently outperforms a purely observational baseline.

A limitation of our method is the variational family we use. The proposal for variational posterior considers that the stick-breaking weights, the intervention embeddings, and intervention targets are independent of each other. In some cases, this can potentially create a surrogate score whose maximum structure is not an element of the $\mathcal{I}$-Markov equivalence class.

There are many avenues for future work. Our framework is particularly appealing for problems where performing interventions explicitly is expensive or unethical, but where interventions occur naturally in the data without being explicitly observed. Experimenting with more flexible variational families also seems appealing, albeit this may come at the cost of the closed-form expressions for the KL divergences. Another direction of future research is to address the case where the dataset does not contain samples from the source SCM (observation distribution). In our formulation, we pin the empty set to the first intervention, i.e., the first intervention SCM is the source SCM, which necessarily assumes that we always have samples from the observation distribution. We can extend our method for cases where this is not the case. Under such circumstances, we can define the observational model as the one corresponding to the union of the parent sets of each particular variable across the different intervention regimes.

## Acknowledgments

This work was supported by the European Research Council (ERC StG DeepSPIN 758969) and by the Fundação para a Ciência e Tecnologia through contract UIDB/50008/2020.

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

## Appendix A. Perfect Interventions

If we assume that the intervention is perfect (stochastic or atomic), the log-probability of single data-point is given by

$$\log p(x|z, \tilde{\mathcal{M}}, \mathcal{G}; \theta) = \sum_{j=1}^{d} \log g_j(x_j | (1 - (e_z^\top \mathcal{R})_j) \left( A_j^{\mathcal{G}} \odot x \right), (e_z^\top \mathcal{R})_j \left( e_z^\top \mathcal{U} - u_0 \right) + u_0; \theta_j),$$

(19)

that is, we make the conditional density of $x_j$ be only dependent on $u_k$ when intervened.

## Appendix B. Hyper-parameter Selection

We searched hyper-parameters for the $\alpha$, $\gamma$, $\xi_{\mathcal{G}}$ and learning rate in a validation set based on log-evidence. The remaining hyper-parameters pertaining to the neural networks architecture were selected based on validation datasets (data and graphs) and were held fixed for all of the experiments. The architecture parameters can be found in Table 3.

| Hyperparameter name | symbol | value |
|---|---|---|
| # hidden units | $d_h$ | 32 |
| # hidden layers | $N$ | 2 |
| weight decay | wd | $10^{-6}$ |
| intervention embedding size | $h$ | 264 |
| # flow hidden layers | $l$ | 2 |
| # flow hidden units | $F$ | 12 |
| Lagrangian parameters | $(\varphi_0, \mu_0, \eta, \delta)$ | $(0.0, 10^{-8}, 2.0, 0.9)$ |

Table 3: Default Hyperparameters for our causal discovery method.

The Lagrangian parameters $(\varphi_0, \mu_0, \eta, \delta)$ are respectively set to $(0.0, 10^{-8}, 2.0, 0.9)$. We fixed the number of iterations of each subproblem to 500. We found this value large enough to reach convergence in the small-scale problems considered in this paper.

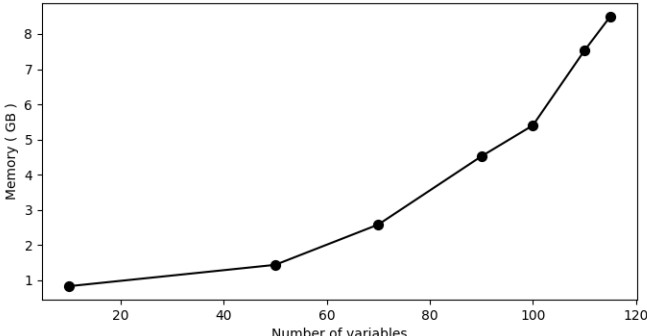

Figure 3: Memory required for inference, using the default hyper-parameters as a function of the number of variables.

## Appendix C. Scalability

In our experiments, we mostly considered problems with a small number of variables, particularly 2, 10, and 20, and just a few number of examples ($10^4$). The main computational bottleneck is the trace exponential matrix operation which is $O(d^3)$ (Buono and Lopez, 2003), where $d$ is the number of variables.

We measured the required GPU memory of our model, as we increasing the number of variables, using the default hyper-parameters and a batch-size of 128, and obtained the plot present in Figure 3. As expected, the memory increases quadratically due to the fact that we represent the graph using an adjacency matrix. Since, we are using mini-batches to estimate the gradients of the loss w.r.t. the parameters of our model the memory requirements stay constant as we scale the number of examples. However, we can expect the convergence time to increase.

## Appendix D. Kullback–Leibler divergences

We detail below the expressions for the Kullback-Leibler (KL) divergences used in this paper.

$$
\begin{aligned}
D_{KL}\big[q(\tilde{\mathcal{M}}, z^{(i)})||p(\tilde{\mathcal{M}}, z^{(i)})\big] &= \\
&= \mathbb{E}_{\tilde{\mathcal{M}}, z \sim q(\tilde{\mathcal{M}}, z)}\left[\log \frac{q(z|u_0,\ldots,u_T)\prod_{k=0}^{T}\Big(\prod_{j=1}^{d} q_R(r_{kj})\Big)\Big(\prod_{l=1}^{h} q_U(u_{kl})\Big)q_V(v_k)}{p(z|\beta_0,\ldots,\beta_T)\prod_{k=0}^{T}\Big(\prod_{j=1}^{d} p(r_{kj}|\gamma)\Big)\Big(\prod_{l=1}^{h} p(u_{kl})\Big)p(v_k|\alpha)}\right] \\
&= \sum_{k=0}^{T}\sum_{l=1}^{h} D_{KL}\big(q(u_{kl})||\mathcal{N}(0,1)\big) + \sum_{k=0}^{T}\sum_{j=1}^{d} D_{KL}\big(q(r_{kj})||\text{Bern}(\gamma)\big) \\
&\quad + \sum_{k=0}^{T} D_{KL}\big(q(v_k)||p_\theta(v_k|\alpha)\big) + \mathbb{E}_{\tilde{\mathcal{M}}, z \sim q(\tilde{\mathcal{M}}, z)}\left[\log \frac{q(z|u_0,\ldots,u_T)}{p(z|\beta_0,\ldots,\beta_T)}\right]
\end{aligned}
$$

$$(20)$$

**Stick-breaking weights**

$$D_{KL}\big(q(v_k)||p_\theta(v_k|\alpha)\big) = \log\Big(\frac{\mathrm{B}\big(\rho_k w_k, (1-\rho_k)w_k\big)}{\mathrm{B}(1,\alpha)}\Big) + (1 - \rho_k w_k)\psi(1) + \\ \big(\alpha - (1-\rho_k)w_k\big)\psi(\alpha) + (-1 + w_k - \alpha)\psi(1+\alpha) \tag{21}$$

**Intervention embeddings**

$$D_{KL}\big(q(u_k)||\mathcal{N}(\mathbf{0}, I_h)\big) = \sum_{j=1}^{h} \frac{\sigma_{kj}^2 + \mu_{kj}^2 - 1}{2} - \log\sigma_{kj} \tag{22}$$

**Intervention targets**

$$D_{KL}\big(q(r_{kj})||\mathrm{Bernoulli}(\gamma)\big) = \pi_{kj}\Big(\mathrm{logit}(\pi_{kj}) - \gamma\Big) + \log\frac{1 - \pi_{kj}}{1 - \sigma(\gamma)} \tag{23}$$

**Correspondence**

$$\mathbb{E}_{\mathcal{V}^k, u_0,\ldots,u_K \sim q(\mathcal{V}^k)q(u_0)\ldots q(u_K)}\Big[ D_{KL}\big(q(z)||p_\theta(z|\beta_0, \beta_1, \ldots, \beta_K)\big)\Big] = \\ = \sum_{k=0}^{K} \mathbb{E}_{u_k \sim q(u_k)}\Big[ q(z_k)\big(\log q(z_k) - \mathbb{E}_{\mathcal{V}^k \sim q(\mathcal{V}^k)}[\log\beta_k]\big)\Big] \tag{24}$$

where

$$\mathbb{E}_{\mathcal{V}^k \sim q(\mathcal{V}^k)}[\log\beta_k] = \mathbb{E}_{v_k \sim q(v_k)}[\log v_k] + \sum_{k'=0}^{k-1} \mathbb{E}_{v_{k'} \sim q(v_{k'})}[\log 1 - v_{k'}] \\ = \psi(\rho_k w_k) + \sum_{k'=0}^{k-1} \psi\big((1-\rho_{k'})w_{k'}\big) - \sum_{k'=0}^{k} \psi(w_{k'}), \tag{25}$$

and $\psi$ is the digamma function. We approximate using the Taylor series expansion, where we use the reparametrization trick on $u_k$ to estimate the expectations.

## Appendix E. Description of the synthetic datasets

To generate the SCMs, we first generated the causal graph. The graphs were generated going in order from node 1 to node $N$ and adding edges from the already visited nodes to the current node $i$ with probability $p$ obtaining the adjacency matrix $A^{\mathcal{G}}$. In order to obtain graphs that weren't topologically ordered we applied a random permutation $P$ to $A^{\mathcal{G}}$. Then, given the causal graph, we sampled the noise distributions and the assignments as described in Table 4. For every SCM, we generate $d$ interventions, one for each variable. How we sample the new assignment for each intervened variables is described in Table 5. Before fitting the model, the data is always normalized. We subtract the mean and divide by the standard deviation.

For each setting, we sample $n/(d+1)$ examples. We used $n = 10000$ in the 10 variable data set. Before using our method, the data is always normalized. We subtract the mean and divide by the standard deviation. For stochastic interventions in variable $j$, we sampled a new Gaussian variable $\varepsilon_j$, as described in Table 5, and assigned it to $x_j$. Using imperfect interventions, we go to the last layer of the neural network ( in the case of linear model its the weight vectors ) representing the assignment, and sample new weights according to Table 5. Table 4 contains a description of how we sample assignments. Particularly, the architecture of the neural network, when it is non-linear SCM. Figure 4 contains scatter plots obtained after sampling the generated two variable SCMs.

| Model Name | $p(\mathcal{E})$ | weights | activation | layers | #hidden units |
|---|---|---|---|---|---|
| Linear Gaussian | $\prod_{j=1}^{d} \mathcal{N}(0, 0.015)$ | $\mathcal{N}(0, 2.0)$ | identity | 1 | - |
| Non-Linear Gaussian | $\prod_{j=1}^{d} \mathcal{N}(0, 0.015)$ | $\mathcal{N}(0, 2.0)$ | relu | 2 | 5 |
| Non-Linear Non-Gaussian | $\prod_{j=1}^{d} \mathcal{N}(0, 0.015)$ | $\mathcal{N}(0, 2.0)$ | relu | 2 | 5 |

Table 4: Description of how assignments are sampled. We set $k \sim \mathcal{N}(0.1, 0.005)$ and $\ell \sim \mathcal{N}(0, 0.4)$.

| Intervention type | new $p(\mathcal{E})$ | new weights |
|---|---|---|
| Atomic | $(1 - 2b) \cdot u$ | 0 |
| Stochastic | $\mathcal{N}((1 - 2b) \cdot u, 0.1)$ | 0 |
| Imperfect | $\mathcal{N}((1 - 2b) \cdot u, 0.1)$ | $\mathcal{N}(0, 2.0)$ |

Table 5: Description of how assignments are sampled for each intervened variable. We set $u \sim \mathcal{U}[1.2, 2.2]$ and $b \sim \mathrm{Bernoulli}(0.5)$

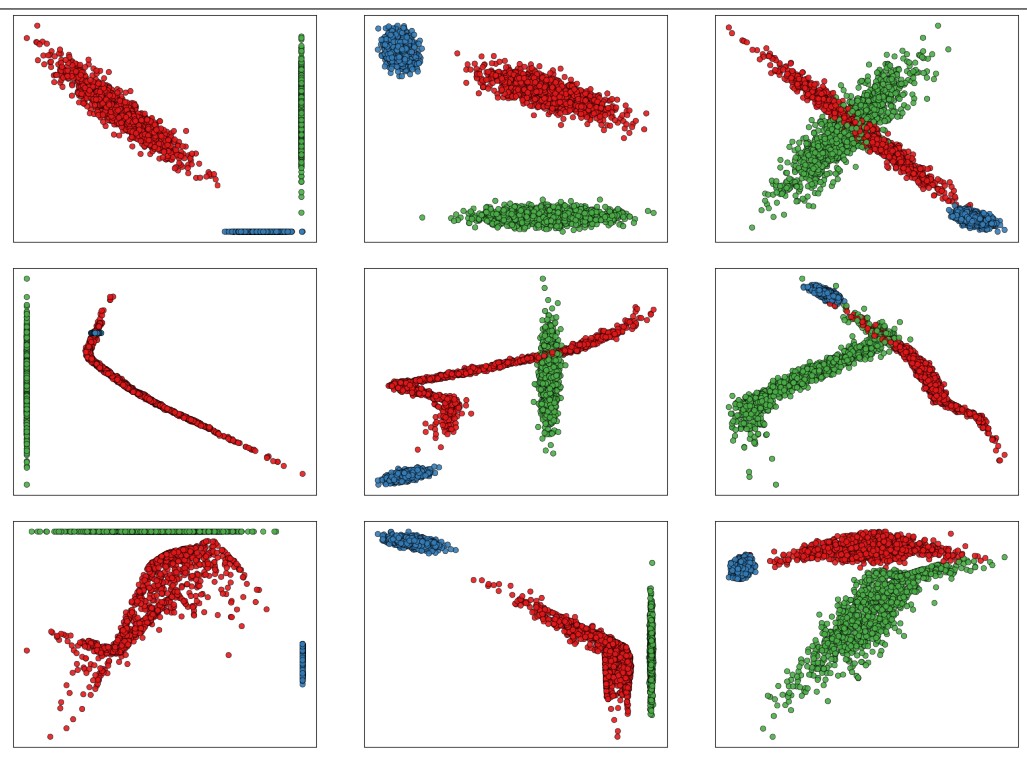

Figure 4: The underlying graph is has two variables and one edge. The cause is in the horizontal axis, the effect is in the vertical axis. The rows correspond to different models, particularly, Linear Gaussian, Non-Linear Gaussian and Non-Linear Non-Gaussian. The columns correspond to distinct types of interventions. Particularly, atomic, stochastic and imperfect. Red is observational. Green is intervention on the effect variable. Blue is intervention on the causal variables.

## Appendix F. Neural network details

The neural network we used is a feed-forward (fully connected) network. Given a vector $x \in \mathbb{R}^{d_{\text{in}}}$, we parameterize with a vector of weights $\theta$ a non-linear mapping FFN : $\mathbb{R}^{d_{\text{in}}} \to \mathbb{R}^{d_{\text{out}}}$. This mapping is a composition of linear transformation interchanged with a non-linear element-wise operation. We use many tricks that have been shown to improve the results with neural network modules, particularly dropout, SiLU activation function, layer normalization and residual connections. We group these techniques together into a sub-layer that form the building block of our relatively more complex neural network models, as shown in Figure 5.

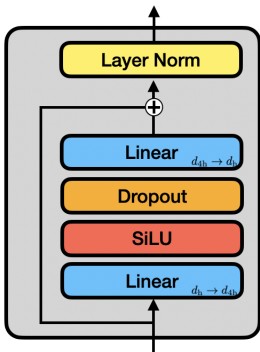

Figure 5: Diagram of the NN-Block sub-layer.

To enable the use of skip connections, we have to make sure that the inputs and outputs of the sub-layer have the same dimension. For this reason before applying the stack of sub-layers we apply a linear transformation from $\mathbb{R}^{d_{\text{in}}}$ to $\mathbb{R}^{d_{\text{h}}}$ and after passing the stack from $\mathbb{R}^{d_{\text{h}}}$ to $\mathbb{R}^{d_{\text{out}}}$, as shown in the diagram in Figure 6.

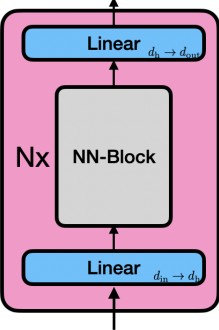

Figure 6: Diagram of the Feed Forward Network (FFN) we use as functional approximator in our model.

## Appendix G. Semi-supervised learning experiments

We test our semi-supervised method, on 10 variable linear SCMs generated according to Appendix E. We consider stochastic and imperfect interventions. We sampled a 10000 samples dataset from each SCM. For each dataset, we created multiple semi-supervised learning datasets by splitting the original 10000 samples into labelled (with intervention assignments) and unlabelled (without intervention assignments) according to some fraction $f$. The fraction $f$ goes from 0 to 1, sampled every interval of 0.2. This means that, for each fraction $f$, we have 10 distinct semi-supervised datasets for both perfect and imperfect interventions. We picked the default hyper-parameters, particularly, $\xi_\mathcal{G} = -0.1$, and $\gamma = -0.01$. The results are contained in Figure 7.

The results suggest that there is no significant improvement from using the correspondences for a fraction of the samples under the tested conditions.

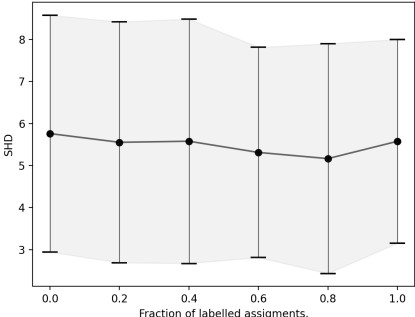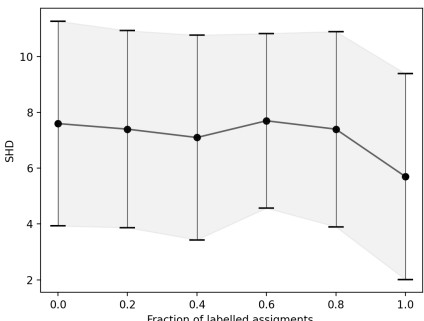

Figure 7: Semi supervised experiments on 10 variable Linear SCMs with an expected value of 1 edge per node. On the left, perfect interventions. On the right, imperfect interventions.

