# OpenReview forum: "Differentiable Causal Discovery Under Latent Interventions"
_cclear.cc/CLeaR/2022/Conference — CLeaR 2022 Poster_

### Official Review · Reviewer_r53E · 2021-11-19

**Confidence:** 4
**Overall Score:** 4

**Main Review:**

The problem addressed is important since the setting considered is realistic and challenging. However, I have a few concerns: 1) since the authors don’t show the consistency of their methods, I would expect the experiments to be more thorough and 2) the method is not well situated within the causal literature, more particularly the link with latent confounders.

It is not clear how this setting is different from causal discovery in the presence of latent confounders. The differences should be clearly explained. If there are no differences, then several other state-of-the-art methods exist (e.g., Spirtes et al., 1999) and, moreover, compared to the proposed method, they often have guarantees to recover the equivalence class. In this case, the authors should definitely include these methods in their related work section, compare their method to at least one of these, and explain clearly how their proposed method differs. To develop the latter point, the related work is really short and doesn't situate the present work in the causal discovery literature.

This work has a limited novelty. It is based directly on Brouillard et al., 2020 that is also a continuous-constrained optimization method that relies on neural networks and can deal with interventional data, possibly with unknown targets (also it uses the same components: random matrices trained with the Gumbel-Softmax trick, normalizing flows (DSF), augmented Lagrangian, etc). The main contribution of the present work is to add variational inference in order to deal with latent interventions. However, contrary to previous works, in this new condition, the authors don’t have any identification results and thus, the graph with the best score can be an element that is not part of the I-Markov equivalence class. As mentioned in the previous paragraph, if their setting is equivalent to the presence of latent confounders, then the novelty is furthermore reduced since other methods exist with theoretical guarantees.

**Experiments**
Since they don’t have theoretical guarantees, I would have expected experiments to be more convincing.

For the synthetic dataset experiments, they don’t have any external baseline method. They show that “latent” is comparable to “known”. I wonder if it is simply explained by the fact that the different environments are easily distinguishable (see Fig. 3, it seems to be the case, more obviously for atomic interventions). If it is the case, then a method that first finds the environments (performing some kind of clustering for example) and then performs classical causal discovery with unknown targets would have a similar performance. This would also explain the results of the semi-supervised learning experiments where the knowledge of the environment doesn’t help.

All the experiments are performed on very small graphs (2-11 vertices). Furthermore, the authors state that their method scales well w.r.t. the number of interventions. It would be interesting to see if this method is applicable to larger graphs and how well it scales in practice.

For the real-world dataset, since their assumptions are less restrictive, I would expect to have at least another dataset than the classical Sachs et al., 2005 (which is imperfect as noted by the authors).

For the hyperparameter selection, the procedure is not clearly explained. How many experiments were performed in order to determine the optimal $\alpha$, h, K, learning rate, etc; and was it done on only one type of dataset? If I understand correctly, once determined, the same hyperparameters are applied for all the experiments and a grid-search was only performed on $\lambda_G$ and $\gamma$? The authors state that the models were trained for 500 epochs for the first iteration of the augmented Lagrangian. What are the parameters used for the augmented Lagragian? Why do the authors use a fixed number of iterations?

For the reproducibility of the experiments, while the authors state that the code will be provided, all the hyperparameters used should be clearly described in the appendix (how many layers for each neural network, the hyperparameters related to the augmented lagrangian, etc).

**Method**
In Section 3.2, if I understand correctly $\lambda_G$ is a scalar defined by the user. Why are all the $\lambda_{ij}$ set to this value? It seems they should be learned independently.

In Section 3.3, instead of using different neural networks for each interventional distribution, the authors use an intervention embedding $\mathcal{U}$ and share the parameters $\theta_j$ between all interventions. By doing this, they have a method that scales well w.r.t. the number of interventions, but they don’t fully use the invariance of the conditional for different interventional settings. Also, it gives them a prior that causal mechanisms should be similar when intervened upon, which is not always the case.

**Typos and short comments**
- In abstract, “a infinite” -> an infinite
- Abstract, extra space after “models”
- Section 2.1, what is S in the SCM?
- Section 2.4, to be correct, Zheng et al., 2018 proposed the constraint $tr(e^{W \odot W}) -d = 0$
- Section 4: “All of the possible graphs share the MEC”. The same MEC? It is not true, the two first graphs share the same MEC (A -- B), while the last one has its own MEC (A  B).
- Section 4: “is worse than”
- Section 4: “with a full batch (” (space before the parenthesis)
- Section 5, p.12: “representation”
- Section 5, real-world dataset: should cite Joris et al., 2020 for the observations related to Sachs.
- Appendix B, what is $k$ and $l$ in the Table 3 caption?
- Appendix C, “Given an vector”
- Appendix C, “as show in Figure 4”
- Appendix D, “a the correspondences”
- Appendix D, there are extra spaces in parentheses
- When citing, usually the publications are in order starting from the most recent to the oldest

**Summary:**

The authors propose a score-based causal discovery method for the setting where interventions are performed latently (i.e., the targets are unknown and even the environments from which they are sampled are unknown). More specifically, the authors propose a continuous-constrained optimization method that relies on neural networks and variational inference. The authors show competitive results compared to the case where the environment is known but the target is unknown.

---

> ### Author Response · Authors · 2021-12-04
> **Response to reviewer r53E**
>
> Thank you for the careful reading and insightful comments. We answer your main points below, we hope that our answers alleviate your concerns.
>
> > “It is not clear how this setting is different from causal discovery in the presence of latent confounders.”
>
> This is a good question. Note, however, that there is a very significant difference, in terms of underlying assumptions, between our method and that of Spirtes et al. (1999). As you mentioned, we propose a score-based causal discovery method for the case where the interventions are **fully** latent: i.e., we do not know what they alter or which samples are affected. Latent interventions are not the same as latent confounders, as the latter are always present, whereas the former are only present in some samples, and we don't know which samples. In our approach, we encode the latent interventions with the variables z (latent correspondences), and r (latent intervention targets), as described in Sec 3.3, using a shared intervention space. While, as you point out, we could regard these two sets of variables as hidden confounders of an “expanded” causal model, treating them in that way would disregard the structure of the problem encoded in z and r, and this structure is key to the success of our approach. In practice, this leads to inferior results, as our new results with the FCI algorithm show (see the response to the next question below). Regarding recovery guarantees of the equivalent class, while the FCI algorithm (Spirtes et al. 1999) and other constraint-based algorithms allow for latent confounders and selection bias, these methods have been designed to estimate the causal graph of the system from a single dataset corresponding to a  purely observational context (not our scenario). In other words, they also do not have guarantees of recovering the I-MEC.
>
> For the complex problem we are tackling, a guarantee of I-MEC seems unlikely for any method, without very strong additional assumptions: it would necessarily require some sort of z-faithfulness property (difficult to ensure in a real-world problem) as well as the usual I-faithfulness and faithfulness. While our method does not have these formal guarantees (which we point out would be very hard to get for any method), it is intuitive and performs well in practice.
>
> > “... the authors should definitely include these methods in their related work section, compare their method to at least one of these, and explain clearly how their proposed method differs. (...) the related work is really short and doesn't situate the present work in the causal discovery literature.”
>
> Thanks for the suggestion, we will expand the related work section and will include a discussion about these methods, explaining the differences with respect to our method. Regarding a comparison, we followed your suggestion and tested FCI (with Fisher z-test with significance level 0.05). We used the package causal-learn (https://github.com/cmu-phil/causal-learn). We obtained the following results, on our linear Gaussian synthetic dataset (n=10, e = 1, stochastic): SHD: 20.8 $\pm$ 4.1 vs SHD: 35 in Sachs et al. (2005), with F1=0.22, tp=4, fn=12, fp=21, rev=5. We will report these results, which are overall inferior to our method. Particularly, for the synthetic dataset, FCI tends to produce significantly denser graphs.
>
> > “All the experiments are performed on very small graphs (2-11 vertices). (...) It would be interesting to see if this method is applicable to larger graphs and how well it scales in practice.”
>
> Good suggestion, we will run experiments on larger graphs and will report them in the final version.
>
> > “For the hyperparameter selection, the procedure is not clearly explained. (...)  If I understand correctly, once determined, the same hyperparameters are applied for all the experiments and a grid-search was only performed on $\lambda_G$ and $\gamma$”.
>
> We searched hyperparameters for the $\alpha$, $\gamma$, $\lambda$ and learning rate in a validation set based on log-evidence. Those pertaining to the architecture of the neural networks were selected based on validation datasets (data + graphs) and were held fixed for all of the experiments (the results were generally robust to them). We will include all the details in the appendix.
>
> > “For the reproducibility of the experiments, while the authors state that the code will be provided, all the hyperparameters used should be clearly described in the appendix (how many layers for each neural network, the hyperparameters related to the augmented lagrangian, etc).”
>
> We agree, and we will follow this suggestion.
>
> > "In Section 3.2, if I understand correctly $\lambda_G$ is a scalar defined by the user. Why are all the $\lambda_{ij}$ set to this value? It seems they should be learned independently."
>
> This $\lambda_G$ is from the prior distribution, not the graphs’ weighted adjacency matrix. We will clarify this by changing the notation.

---

### Official Review · Reviewer_XPfV · 2021-11-23

**Confidence:** 2
**Overall Score:** 6

**Main Review:**

The paper focuses on the under-explored problem of score-based causal discovery, where for each sample we don't know which was the interventional distribution that generated it. It proposes a differentiable structure learning approach based on variational inference and Dirichlet processes. The framework can be extend to the case of partially knowing the intervention targets, or at least knowing which batch of samples is generated by the same intervention.

The paper is generally well-written and to the best of my knowledge technically sound. I thought the setting was particularly interesting, with many applications.

One of my concerns is that results are only on single-node interventions on simulated data (and with single node interventions on each of the variables) and the [Sachs et al. 2005] dataset, which is not the best benchmark for this case. I would really be curious about any result on multiple-targets and in the case in which one doesn't have an intervention per variable. Even in this case, the stochastic interventions results are a bit lukewarm, and I'm confused why there are no perfect interventions.


# Minor points

Currently the method assumes the parents cannot change in an intervention - would it then make sense to just consider as the "original" SCM M the one in which you can have the union of the sets of parents (assuming it is still acyclic) in the different environments and then switch them off in each specific environment?

If the $\theta_j$ are shared across interventions, I'm not sure I understand exactly how the soft interventions are modelled


**Summary:**

Interesting problem, sound approach, empirical evaluation should be improved

---

> ### Author Response · Authors · 2021-12-04
> **Response to reviewer XPfV**
>
> Thank you for the careful reading and overall positive comments about our paper.
>
> Please note that our method assumes that, in an intervention, the parent set is always a subset of the parent set in the observational graph; consequently, the parent set does change, but it needs to be  a subset of the original set of parents. As the reviewer hints, we can indeed extend our method to define the observational model as the one corresponding to the union of the parent sets (though this needs to be done in all variable nodes simultaneously) even if we never observe samples from it. We will mention this possibility in the final version; we thank the reviewer for pointing it out!
>
> Concerning the $\theta_j$ being shared across interventions, this is actually one of our core contributions. We have a meta-model for all of the assignments (comprising the effects of the potentially infinite set of interventions) of each variable j.  To select a particular assignment k, we define an intervention embedding ($u_k$, see Section 3.3) on which we condition on. In this way, the parameters $\theta_j$ are shared among the  interventions -- they control the meta model, and not specifically each intervention assignment function. Each assignment function therefore depends on both $\theta_j$ and $u_k$ for intervention $k$. This is the reason why  we propose the intervention embeddings and the Dirichlet process formulation. We will make this clearer in the final version.
>
> Thanks for the suggestions regarding the empirical evaluation, we will add results to additional experiments to the appendix. Note that our stochastic interventions are a particular case of perfect interventions as defined by Brouillard et al. 2020: there we remove the dependencies from the parents, and define a unary distribution for the node (rather than an atomic value). We will run experiments on the  flow cytometry dataset using our model with perfect/stochastic interventions as well.

---

### Official Review · Reviewer_gF7i · 2021-11-24

**Confidence:** 3
**Overall Score:** 9

**Main Review:**

# Quality and clarity
- This paper is well written, well organized, and clear.
- It contains a theoretical reasoning of their method, and their method is likely reproducible.
- The paper is self-contained, with a good description of previous studies and existing methods.
- The paper contains the results of the experiments. Some of the experimental results of the proposed method are inferior to those of the existing methods, but reasonable explanations are given for the results.

# Originality and significance
-  This paper extends Brouillard et al. (2020) by relaxing their assumptions. Using interventional data for causal discovery is a very important problem but little has been studied. The relaxing of the assumption is important for the causal discovery from interventional data.

# Feedback to the authors
- What variable does the S in the first line of Section 2.1 refer to?

**Summary:**

This paper proposes a causal discovery method using leveraging interventional data. This study extends Brouillard et al. (2020). The previous study assumes that the correspondence between samples and interventions is known. This paper relaxes the assumption and propose a method based on neural networks and variational inference.

---

> ### Author Response · Authors · 2021-12-04
> **Response to reviewer gF7i**
>
> Thank you for your positive comments about our paper.
>
> > "What variable does the S in the first line of Section 2.1 refer to?"
>
> Good catch; S refers to the set of assignments in Eq. 1. We will clarify this in the final version.

---

### Decision · Program_Chairs · 2022-01-12

**Decision:**

Accept (Poster)

**Comment:**

I recommend acceptance and editing the manuscript to highlight the limited nature of experiments (more datasets and more realistic applications would be welcome). Still, all reviewers see this submission as interesting and worth presenting in the conference.